# Asian elephants (*Elephas maximus*) reassure others in distress

Joshua M. Plotnik and Frans B.M. de Waal

Living Links, Yerkes National Primate Research Center and Department of Psychology, Emory University, Atlanta, GA, USA

## ABSTRACT

Contact directed by uninvolved bystanders toward others in distress, often termed consolation, is uncommon in the animal kingdom, thus far only demonstrated in the great apes, canines, and corvids. Whereas the typical agonistic context of such contact is relatively rare within natural elephant families, other causes of distress may trigger similar, other-regarding responses. In a study carried out at an elephant camp in Thailand, we found that elephants affiliated significantly more with other individuals through directed, physical contact and vocal communication following a distress event than in control periods. In addition, bystanders affiliated with each other, and matched the behavior and emotional state of the first distressed individual, suggesting emotional contagion. The initial distress responses were overwhelmingly directed toward ambiguous stimuli, thus making it difficult to determine if bystanders reacted to the distressed individual or showed a delayed response to the same stimulus. Nonetheless, the directionality of the contacts and their nature strongly suggest attention toward the emotional states of conspecifics. The elephants' behavior is therefore best classified with similar consolation responses by apes, possibly based on convergent evolution of empathic capacities.

Corresponding author
Joshua M. Plotnik,
Joshua.Plotnik@gmail.com

## INTRODUCTION

Most empirical evidence for how animals react to others in distress comes from the study of conflict resolution (*de Waal & van Roosmalen, 1979*; *de Waal & Aureli, 1996*; *de Waal, 2000*). Peacekeeping mechanisms have evolved to manage conflict in animal societies (see *de Waal & Aureli, 1996*; *de Waal, 2000*, for a review), including reconciliation (i.e., affiliative physical contact between former opponents soon after a conflict) and consolation (i.e., affiliative physical contact from an uninvolved bystander directed toward a recipient of aggression). The former is much more common than the latter in the animal kingdom, possibly due to differences in the complexity of underlying cognitive mechanisms (*de Waal & Aureli, 1996*; *de Waal, 2008*). Although reconciliation appears to be self-interested for all individuals involved due to the need to maintain valuable relationships, the significance of consolation for the bystander is still unclear (*de Waal, 2000*). Recent work trying to identify the adaptive function(s) of consolation has focused on (a) the identity of bystanders and their relationships with the consolation recipient (*Romero, Castellanos & de Waal, 2010*; *Romero & de Waal, 2010*; *Romero, Castellanos & de*

*Waal, 2011*), (b) the physiological changes in distressed individuals consoled by bystanders (*Koski & Sterck, 2007*; *Fraser, Stahl & Aureli, 2008*), and (c) possible benefits to the consolers themselves (*Koski & Sterck, 2007*; *Koski & Sterck, 2009*). All of these possible functions suggest that the parties involved initiate or accept contact as a way of mitigating emotional stress responses (*de Waal, 2008*; *Koole, 2009*).

Because of these functional uncertainties, some scientists remain reluctant to use functional or motivational terminology, such as consolation; instead, the aforementioned behavior is sometimes described as "third-party affiliation" (a descriptive term that specifies only directed, physical contact with a distressed individual, e.g., *Call, Aureli & de Waal, 2002*; *Koski & Sterck, 2007*; *Seed, Clayton & Emery, 2007*). However, other studies argue that the mammalian capacity for empathy underlies consolation (*Preston & de Waal, 2002*), and compare the morphology and motivation of the behavior with "sympathetic concern" in humans (*Romero, Castellanos & de Waal, 2010*; *Clay & de Waal, 2013*). In general, demonstrations of consolation in animals are rare, with empirical evidence thus far provided only for the great apes, canines, and certain corvids (*de Waal & van Roosmalen, 1979*; *de Waal & Aureli, 1996*; *Palagi, Paoli & Tarli, 2004*; *Cordoni, Palagi & Borgognini Tarli, 2006*; *Mallavarapu et al., 2006*; *Seed, Clayton & Emery, 2007*; *Cools, van Hout & Nelissen, 2008*; *Palagi & Cordoni, 2009*; *Fraser & Bugnyar, 2010*; *Romero, Castellanos & de Waal, 2010*; *Romero & de Waal, 2010*), but not for monkeys or any other species (e.g., *de Waal & Aureli, 1996*; *Schino et al., 2004*; *Watts, Colmenares & Arnold, 2000*, but see *Call, Aureli & de Waal, 2002*; *Wittig et al., 2007* for examples of comparable affiliative behavior). This rarity may be due to the potential cognitive underpinnings of consolation, such as empathic perspective-taking (*de Waal, 2008*), or else to species-specific social dynamics that determine how animals mitigate social strife in a variety of relationships. In some monkey societies, for example, it may be too risky to associate with victims of aggression due to the strictness of their linear hierarchies (*de Waal & Aureli, 1996*; *de Waal, 2000*).

Elephants are an interesting study species because of their complex social behavior and close bonding with family members (*Douglas-Hamilton & Douglas-Hamilton, 1975*; *Moss, 1988*; *Poole, 1996*; *Schulte, 2000*; *Payne, 2003*; *Bates et al., 2008*). They often act as allomothers toward others' offspring, and respond immediately to the vocalizations of these individuals (e.g., in response to infant distress – *Lee, 1987*, *Bates et al., 2008*). They are also known for their "targeted helping," or directed assistance that takes the specific needs of others into account (e.g., helping to lift and coordinated bracing of injured, dying or otherwise prostrate family members – *Douglas-Hamilton et al., 2006*; *Bates et al., 2008*). Targeted helping is viewed as a sign of empathic perspective-taking (e.g., *Preston & de Waal, 2002*; *de Waal, 2008*).

In the present study, we aim to assess the affiliative tendencies of Asian elephants (*Elephas maximus*) toward conspecifics in response to distress, using similar methodology to that used in the conflict resolution literature. To our knowledge, this is the first systematic investigation of distress-related affiliation in elephants based on *a*

*priori* hypotheses (but see *Bates et al., 2008* and *Hart, Hart & Pinter-Wollman, 2008* for other possible displays of empathy and stress-related emotional responses).

Relative to chimpanzees (*de Waal, 1982*; *de Waal & Aureli, 1996*), elephants do not often engage in conflict within their herd, which consists primarily of related females and immature offspring (*Douglas-Hamilton & Douglas-Hamilton, 1975*; *Moss, 1988*; *Poole, 1996*; *Payne, 2003*; *de Silva, Ranjeewa & Kryazhimskiy, 2011*). Thus, we measured how elephants affiliate or reassure others as a response to an individual's distress irrespective of its cause. We recognize that our inability to identify a clear stimulus for each distress event makes it difficult to differentiate between cases where individuals are reacting directly to the stimulus or to another elephant's distress. Because of this, it is unclear if all or most cases of affiliative contact can be classified as "consolation" in the way this label is used in other post-conflict studies (e.g., *Call, Aureli & de Waal, 2002*; *Preston & de Waal, 2002*; *Koski & Sterck, 2007*; *Seed, Clayton & Emery, 2007*; *Cools, van Hout & Nelissen, 2008*; *Fraser, Stahl & Aureli, 2008*; *Koole, 2009*; *Koski & Sterck, 2009*; *Fraser & Bugnyar, 2010*; *Romero, Castellanos & de Waal, 2011*). Instead, we refer to the elephants' affiliation with others as "reassurance" to note our focus on both affiliative contacts and emotional responses. We use this term instead of "consolation" to avoid implying the potential function of the elephants' behavior.

This study tries to distinguish the affiliative tendencies of elephants in response to behaviorally identified stress. Based on the aforementioned social complexity of and targeted helping in elephants, we predicted that reassurance behavior toward distressed individuals should be identifiable through an assessment of physical and vocal contacts. If elephants are responsive to the distress of others, they should be expected to make physical or vocal contact with stressed conspecifics, and do so sooner than in control periods during which the conspecifics do not display distress. In addition, we might expect emotional contagion – bystanders' adoption of the emotional state of those in distress - to be part of such a reaction if the elephants' affiliative behavior is part of a more complex, emotionally driven social response (*Zahn-Waxler, Hollenbeck & Radke-Yarrow, 1984*; *Zahn-Waxler & Radke-Yarrow, 1990*; *Zahn-Waxler, Hollenbeck & Radke-Yarrow, 1984*; *de Waal, 2003*, *2008*; *Clay & de Waal, 2013*). Thus, we predicted that the elephants' behavioral and emotional responses would mimic physically and follow temporally those of distressed conspecifics. Matriarchal elephant herds exhibit close social bonding and often display varying levels of emotional reactivity (e.g., *Moss, 1988*; *Poole, 1996*; *Schulte, 2000*; *Payne, 2003*; *Bates et al., 2008*). Because of this, we also considered that emotional contagion, found in many mammals (see *de Waal, 2003*, *2008*), might lead to affiliative interactions among bystanders as well. Thus, we also predicted that bystanders to distress would make physical or vocal contact with one another, in addition to, or instead of contact with the first stressed individual.

## MATERIALS AND METHODS

### (a) Study area and subjects

This study was conducted at the Elephant Nature Park (the "Park") in the Mae Tang district of Chiang Mai province, Thailand. Although the Park owns many of the elephants

on-site, some are leased or contracted so that the general elephant population changed regularly during the study period. The data in this study refer to 26 elephants with approximate ages ranging from 3 to 60 years old, although due to unverifiable records, ageing elephants precisely was impossible. Although genetic tests on the relatedness of the elephants were never done, it is reasonable to conclude based on the relayed life histories of the individual elephants that all individuals, except for mother-juvenile pairs brought to the Park together, were unrelated. Each elephant was taken care of by one or two mahouts (elephant caretakers) every day. Adult male elephants ($n = 4$) were completely excluded from the study as they were regularly prevented, for safety and husbandry reasons, from participating in most of the natural, social interactions within groups. When a female was first brought to the Park, she was generally allowed to integrate with a smaller group of elephants. In this study, these smaller, social units (generally of $n = 5–7$ individuals) are labeled "managed groups" because they consisted of individuals that spent most of their social time together under the guidance of their mahouts. There was no single herd at the Park, but six individual managed groups that interacted at specific times during the day. These groups were delineated based on interviews with the Park mahouts during data collection but prior to data analysis.

Each day, elephants followed a specific routine established by Park management. Mahouts moved their elephants to a specific location on the property, as a managed group, beginning at 0700 h. They ate at a central location at 1130 h – fed either by their mahouts or visiting tourists – bathed communally at 1300 h and 1630 h, and returned to their night shelters, in which they were tethered for the night, at 1700 h. Mahouts moved elephants with vocal commands or by grasping their ears or legs and walking them to different locations on the property. Throughout the day, elephants were left to graze or play in various parts of the property within their social groups. Although individual elephants were generally allowed to interact with members of other managed groups, the mahouts often intervened at unpredictable times to separate volatile pairings.

## (b) Defining distress

Because there is very little literature on Asian elephant behavior in general (but see *Sukumar, 2003*; *Sukumar, 2006*; *de Silva, Ranjeewa & Kryazhimskiy, 2011*), the more detailed literature on African elephant behavior (*Loxodonta* genus – e.g., *Douglas-Hamilton & Douglas-Hamilton, 1975*; *Moss, 1988*; *Poole, 1996*; *Payne, 2003*) is often applied to Asian elephants as well because of their relatively close phylogenetic proximity (*Payne, 2003*). *Douglas-Hamilton & Douglas-Hamilton (1975)* and *Lee (1987)* describe distress in individual elephants, specifically infants, based on specific vocalizations and stimuli. Infants give a specific call – either an infant roar or squeal – and assume an alert posture where the head is raised, the ears are extended, the tail is raised and the trunk is either raised or stiffened outward (*Olson, 2004*). Roars, rumbles, and trumpets are often given in response to infant distress calls, or as a signal of an adult's own distress. Using (1) *Lee*'s (*1987*) definition of distress events in calves as those that result in "a dramatic response on the part of other animals... rushing to assist the calf" (p. 287), (2) *Bates et al.*'s (*2008*) definition of empathic responses to distress as: "A

voluntary, active response to another individual's current or imminent distress or danger, that actually or potentially reduces that distress or danger" (p. 208), and (3) a comprehensive ethogram of elephant behavior with specific attention to those behaviors occurring when an infant or adult is distressed or agitated (adapted and expanded from *Olson, 2004*), we define a distress event in elephants as follows:

> *A distress event is one resulting from an unseen or seen negative stimulus (e.g., mahout-driven separation or movement of individuals, conspecific intimidation or aggression, group separation, environmental threat or accident) that causes an individual to become agitated and to signal such agitation to others (which can be visually identified with specific changes in body state – ears forward, tail erect – and movement, and acoustically identified by various vocalizations, specifically trumpets, roars and rumbles).*

### (c) General data collection

We chose locations on the property from which to collect data to ensure both a full view of pre-selected managed groups and the observer's safety. These locations included viewing platforms constructed specifically for observations, and in fields in close proximity to mahouts. Observation locations were chosen based on three factors in decreasing priority: (1) safety of observation vantage point at any given time, (2) view of a maximum number of managed groups at the beginning of the observation period, and (3) view of the managed groups from which there were the least amount of data. The property was approximately 55 acres in total size, but only 30 acres were observable for this study. The property was divided into four grids for observation purposes, and an observation location was chosen within a grid based on the aforementioned factors.

On average, data were collected during 1–2 week periods each month from April, 2008–February, 2009. General observation periods ran for no less than 30 min and no more than 180 min per session from 0730 to 1030 and from 1400 to 1630, with scan samples taken every 10 min. Data on proximity distance only were collected for relationship quality within elephant groups. All observation periods began after 10 min of no mahout interference on elephant behavior, and individual scan samples were cancelled if such interference occurred within a given 10-min period. All-occurrence sampling was used for distress behaviors and the reactions of others to these behaviors (*Altmann, 1974*). In addition, if an interaction was clearly and completely observed outside these specific observation periods, the same data were collected ad-libitum (<20% of total cases), and a subsequent control observation period (see below) was scheduled.

### (d) Post-distress data collection – PD and MC observations

Although the human staff responsible for the elephants' care artificially constructed the managed group over several years, we focused on spontaneous, affiliative behavior reflective of natural, social interactions (*de Waal, 1982*; *Sukumar, 2003*, *2006*; *de Silva, Ranjeewa & Kryazhimskiy, 2011*). Post-distress data for this study were collected at the Park on 26 semi-free ranging individuals in six managed groups following the PC (post-conflict, or PD, post-distress)/MC (matched-control) methodology developed for reconciliation and consolation behavior in primates (*de Waal & Yoshihara, 1983*).

The PD period was an observation period in which all approach and affiliative behavior was recorded (as were all data on potential stimuli for distress, individuals present within 50 m, and date, time and weather), for a 10-min block following the first distress display. We chose a 10-min duration because it (a) follows the methodology employed with many other non-human species (see *Aureli & de Waal, 2000*, for a review), and (b) far exceeded the average time of first bystander response to another's distress in a baseline observation period conducted by the first author prior to the start of data collection. Because elephant interactions may involve multiple distressed individuals (*Lee, 1987*), the first individual to vocalize, or display a distress behavior was labeled the victim and thus the focal individual in each PD period. If more than one individual responded simultaneously, the rarest case (if known, the least-often distressed individual) was chosen for observation. Each PD period was compared to an MC (matched control) period, or another 10-min block of observation taken of the victim and bystanders on the next possible day following the PD. An MC period was selected when as many variables from the PD – in prioritized order: high percentage of original individuals present, location, time of day and weather – could be maintained, and, most importantly, no new distress event occurred in the 30 min prior to (or during) the period of observation. An MC was collected within seven days of its corresponding PD (in 80% of PD/MC cases, the MC was collected within 48 h). If an MC was conducted when an elephant that had made contact with the distressed individual in the corresponding PD was absent or more than 25 m away, that PD/MC dyad was excluded from the analysis to avoid biasing the data in favor of our predictions.

## (e) Scan-sampling for proximity – "friends" and "non-friends"

We attempted to differentiate between contact directed toward "friends" (closely-bonded individuals) and "non-friends" (those outside managed groups) by collecting 68 h of scan-sampling proximity data (for procedure, see *Romero & de Waal, 2010*). Although mahouts did not interfere with most day-to-day social interactions within established, managed groups (and thus we were able to specify controlled parameters for the PD/MC data), they often discriminately prevented outsider elephants from coming too close to avoid potential conflict. Such conflict between elephants at the Park was also not representative of natural, wild elephant groups (in which conflict is relatively rare), probably due to a high level of unrelatedness within and between managed groups at the Park. Unfortunately, we were forced to exclude the scan-sampling data from our analysis due to circumstances beyond our control. Thus, we were unable to measure relationship quality and its effect on levels of affiliative behavior in this study.

## (f) Analysis

We used Wilcoxon signed-ranks tests (two-tailed) to analyze the differences between PD and MC pairs because of the relatively small sample size. The data were analyzed by focal individual to avoid biasing the data toward any particularly well-represented focal elephant. In addition, the McNemar test was used to assess the presence or absence of elephant bunching behavior within PD/MC observations (*Siegel & Castellan, 1988*). All tests were two-tailed, and P-values were compared to an alpha level of $\alpha = 0.05$.
## RESULTS

### (a) Physical affiliation following distress

To assess reassurance behavior, we first recorded the timing of the first affiliative interaction between the victim (the first individual in a group to display distress behavior, i.e., vocalizations and body state changes signaling emotional distress or agitation) and any bystander(s), with physical contact and affiliative vocalizations analyzed separately. These data were collected during the 10-min PD period and then compared to the timing of the first affiliative interaction in the corresponding MC period. Following standard procedures developed in primate studies (e.g., *de Waal & van Roosmalen, 1979*; *de Waal & Yoshihara, 1983*; *de Waal & Aureli, 1996*; *Romero & de Waal, 2010*), PD/MC pairings were split into three categories: attracted (pairings in which the first affiliative contact occurred earlier in the PD than in the MC, or no contact occurred in the MC following contact in the PD), dispersed (contact occurred earlier in the MC than in the PD or not at all in the PD), and neutral (affiliative contact times did not differ in the PD and its corresponding MC, or no contact occurred in either) (e.g., *de Waal & Yoshihara, 1983*; *Veenema, Das & Aureli, 1994*; *Verbeek & de Waal, 1997*; *Aureli & de Waal, 2000*).

There were 84 PD/MC observations (and thus 84 distinct initial instances of distress signals) across 18 different focal individuals (mean number of PD/MC observations per individual = 9.5, range = 1–38). Within the 84 PD/MC observations, there were a total of 183 focal-bystander dyads, 171 of which involved at least one affiliative physical contact (e.g., Fig. 1) during the PD period (93.4%). 53 of the 84 PD/MC observations included affiliative contact by multiple individuals directed toward a single focal individual. 12 of the 84 observations were the result of an identifiable stimulus for distress – either directed aggression or a feature in the environment (e.g., helicopter, human or dog in close proximity) – that caused distress first in a single individual. The sample size did not allow for further analysis by stimulus type. In our analysis of affiliative contacts, we were concerned only with the first contacts between bystanders and the focal individual in each of the 84 PD/MC observations. The majority of affiliative contacts occurred within the first min following distress (Fig. 2; see Movie S1 for an example of affiliative contact), and a Wilcoxon signed-ranks test performed on the data by focal individual showed that the difference in frequency of these contacts per individual subject in the first min of the PD (mean ± SD = 7.50 ± 8.49) versus the MC (mean ± SD = 0.44 ± 0.86) was significant ($Z = 3.56$, $n = 18$, $P < 0.001$).

We categorized attracted and dispersed pairs based on whether or not each interaction was "solicited" (the focal, distressed individual approached a bystander to seek reassurance) or "unsolicited" (a bystander was the first to approach the focal, which is sometimes called "true consolation" in primate studies – *Call, Aureli & de Waal, 2002*; *Koski & Sterck, 2007*). When the first affiliative contacts between the focal individual and bystanders in each of the 84 PD/MC observations were analyzed (the usual first step in assessing consolation data – e.g., *de Waal & van Roosmalen, 1979*; *de Waal & Yoshihara, 1983*; *de Waal & Aureli, 1996*; *Romero & de Waal, 2010*), a significant difference was

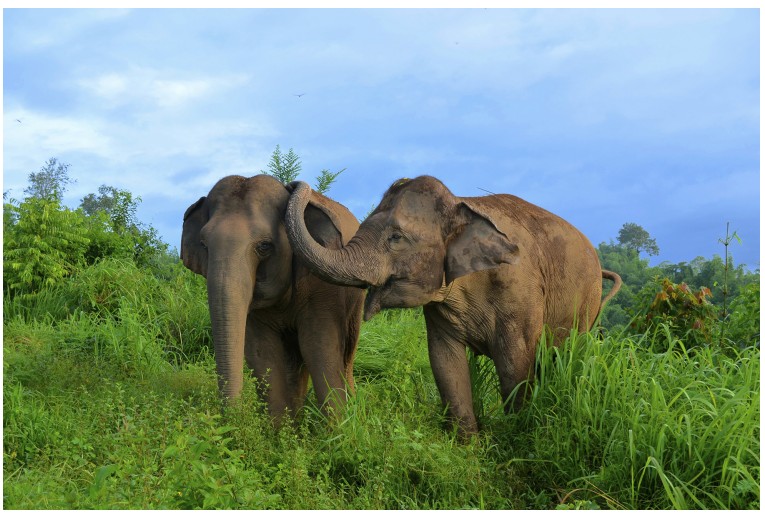

**Figure 1 Physical contact between elephants following distress included trunk touches to the genitals, mouth and the rest of the head (as seen here).** Photograph taken by E. Gilchrist at the Golden Triangle Asian Elephant Foundation, Chiang Rai, Thailand.

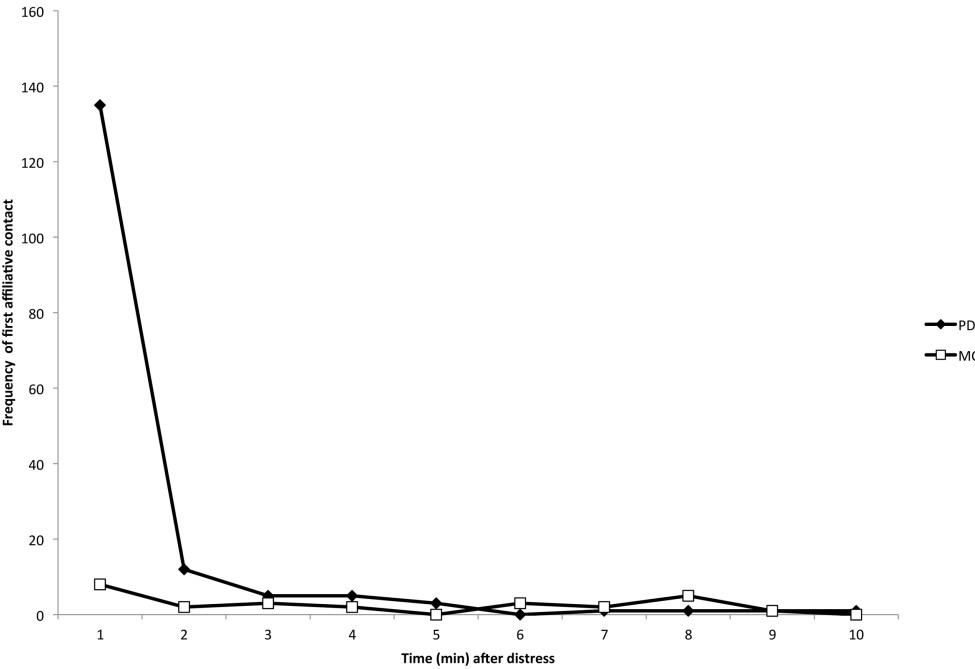

**Figure 2 Temporal distribution of the first affiliative, physical contacts in PD (closed diamonds) and MC (open squares) periods across all dyads.** The number of first contacts occurred overwhelmingly in the first minute following the distress signal, which is consistent with consolation studies in other species (*Aureli, van Schaik & van Hooff, 1989*). See Movie S1 for an example of physical and vocal contact.

**Table 1 Solicited and unsolicited affiliative contacts for each focal individual across all 183 focal-bystander dyads, within 84 PD/MC observations.** Columns indicate numbers of attracted (A), dispersed (D) and neutral (N) pairs per individual in both solicited (S) and unsolicited (US) contacts. Totals are provided in italics. The Mean ± SD indicates the mean proportion of attracted and dispersed pairs ± the standard deviation.

| Focal | A (S) | D (S) | N (S) | A (US) | D (US) | N (US) |
|---|---|---|---|---|---|---|
| AU | 3 | 0 | 0 | 11 | 1 | 1 |
| BT | 3 | 0 | 0 | 7 | 0 | 1 |
| F | 2 | 0 | 0 | 12 | 0 | 1 |
| JB | 0 | 0 | 0 | 2 | 0 | 0 |
| JK | 0 | 0 | 0 | 26 | 0 | 2 |
| MB | 2 | 0 | 0 | 11 | 0 | 0 |
| MD | 0 | 0 | 0 | 3 | 0 | 0 |
| MEL | 3 | 0 | 0 | 0 | 0 | 0 |
| MK | 0 | 0 | 0 | 2 | 0 | 0 |
| ML | 1 | 0 | 0 | 0 | 0 | 1 |
| MLT | 0 | 0 | 0 | 2 | 0 | 0 |
| MP | 6 | 0 | 0 | 1 | 0 | 0 |
| MTK | 3 | 0 | 0 | 0 | 0 | 0 |
| MVL | 0 | 0 | 0 | 3 | 1 | 0 |
| SB | 0 | 0 | 0 | 0 | 0 | 1 |
| TD | 0 | 0 | 0 | 2 | 0 | 0 |
| TJ | 1 | 0 | 0 | 33 | 1 | 9 |
| TT | 0 | 0 | 0 | 19 | 4 | 2 |
| *Group* | *24* | *0* | *0* | *134* | *7* | *18* |
| Mean ± SD | 100% ± 0 | | | 80.31% ± 32.71 | 3.19% ± 7.23 | |

found between the proportion of attracted and dispersed pairs in both unsolicited ($Z = 3.31$, $n = 18$, $P < 0.001$) and solicited contacts ($Z = 2.69$, $n = 18$, $P = 0.007$; Table 1). Across the 18 focal individuals, unsolicited contacts (mean ± SD = 8.83 ± 11.93) occurred significantly more often than solicited contacts (mean ± SD = 1.33 ± 1.71; $Z = 2.47$, $n = 18$, $P = 0.014$). The two most prevalent types of physical contact given by bystanders were trunk touches to another individual's genitals (38.6% of touches), and trunk touches around or inside another's mouth (35.1%; Fig. 3).

### (b) Vocal affiliation following distress

Because elephants primarily use acoustic modalities for communication (e.g., *Poole, 1996*; *Payne, 2003*; *Nair et al., 2009*; *de Silva, 2010*), we also looked at bystanders' vocalizations in response to distressed individuals. In a comparison of first bystander vocalizations in the PD and MC periods, we found that bystanders vocalized earlier following distress than in control periods in a significant number of PD/MC observations (proportion of attracted pairs: mean ± SD = 97.11% ± 8.81%; dispersed pairs: 2.22% ± 8.61%) across 18 focal individuals (incidentally, only three of these focal individuals never had a bystander vocalize when they were distressed: $Z = 3.42$, $N = 18$, $P < 0.001$). Bystander elephants most often chirped (a vocalization often emitted when individuals are in close-proximity to one another – 31.8% of vocalizations) or audibly trunk-bounced (interpreted as a sign of agitation or distress – 24.7% of vocalizations) following distress signals from the focal animal (*Olson, 2004*; *Nair et al., 2009*; *de Silva, 2010*, see Fig. 3).

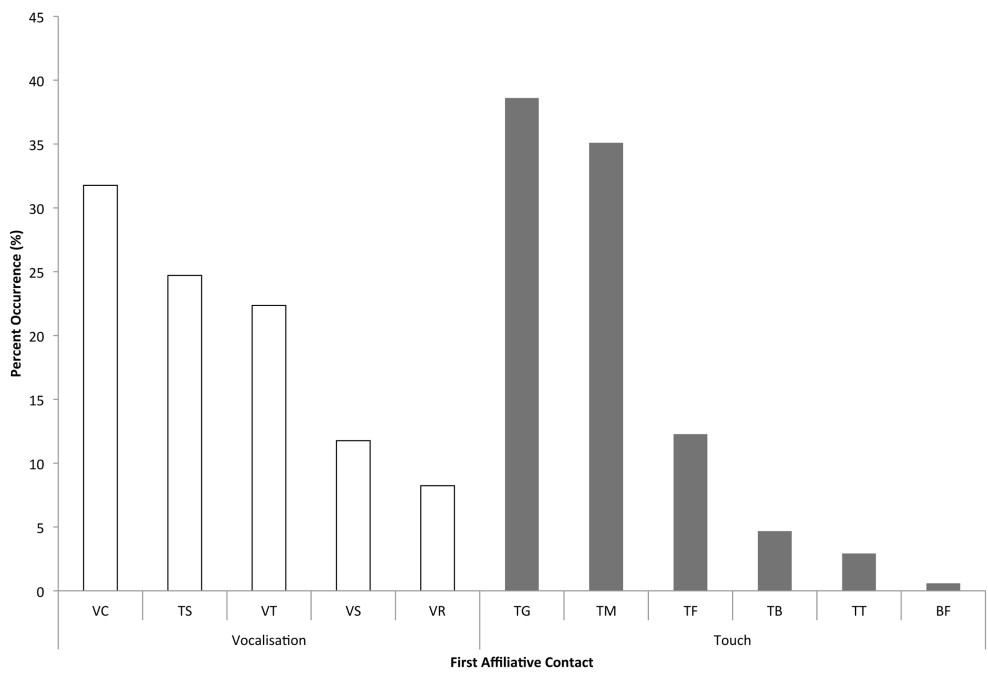

**Figure 3** **Frequency of each type of first contact or bystander response.** Vocalizations: VC - chirp, TS - trunk smack or trunk bounce, VT - trumpet, VS - roar, VR - rumble. Touches: TG - genitals, TM - mouth, TF - rest of face/head, TB - rest of body, TT - trunk/trunk, BF - breast-feeding. The $y$-axis indicates the percent (%) occurrence of each type of vocalization or trunk touch as the first affiliative contact or response across all dyads.

In addition, we assessed differences in the behavior of bystanders in relation to the behavior of distressed individuals between PD and MC periods. Vocalizations may signal agitation or excitement in elephants and are usually paired with similarly functioning physical and postural displays (*Olson, 2004*; *Nair et al., 2009*; *de Silva, 2010*). Bystanders adopted the agitated behavior of the originally distressed focal individual in the PD (e.g., ears presented forward with an erect tail, usually followed by several vocalizations and sometimes with simultaneous urination and defecation – *Olson, 2004*; *Bates et al., 2008*), yet showed no such signs of agitation or distress in the MC in 157 of the 171 dyads in which physical contact occurred (91.8%: mean $\pm$ SD $= 8.72 \pm 9.51$ dyads per focal individual; $Z = 3.56$, $n = 18$, $P < 0.001$).

## (c) Behavior among bystanders
The previous results refer to contact by bystanders to a distressed, focal individual, but we also analyzed contact between bystanders in PDs in which there were multiple individuals present. Bystander-bystander physical contact occurred in 37 of the 84 PD periods, and, like in victim-bystander contacts, occurred earlier following the victim's distress in the PD period than in the MC in a significant number of interactions across 19 possible bystanders (proportion of attracted pairs: mean $\pm$ SD $= 97.37\% \pm 8.36\%$; dispersed pairs: $2.63\% \pm 8.36\%$; $Z = 3.85$, $n = 19$, $P < 0.001$).

Elephants may quickly form a close circle, known as "bunching," around their young in anti-predator defense (e.g., *Moss, 1988*; *Poole, 1996*; *McComb et al., 2001*; *Bates et al., 2008*). Bunching involves the coming together of multiple individuals around the distressed elephant so that all individuals are within trunk's reach of one another (*Nair et al., 2009*). To systematically assess whether individual signs of distress trigger such behavior, we looked at the occurrence/non-occurrence of bunching in PD/MC observations. We excluded all observations in which less than four individuals were present (this excluded $n = 7$ focal individuals altogether). In 30 of the 42 qualifying PD/MC observations, bunching around both juveniles and other adults occurred following distress and never in the corresponding control periods (McNemar change test comparing presence or absence of bunching in PD and control periods: $\chi^2 = 28.03$, df $= 1$, $P < 0.001$).

## DISCUSSION

In this study, we set out to investigate the affiliative responses of elephants to others and found that they engage in more such responses following distress than during control periods. The elephants engaged significantly more often in unsolicited physical contacts (bystanders approached and affiliated with the first-distressed individual) than in solicited contacts (the first-distressed individual is the initiator of the contact). Bystanders also vocalized toward or in response to distressed individuals, and made contact with each other significantly more often than in controls.

In the study of consolation in animals, the stimulus event is almost always a conflict, and the roles of the individual participants – victim, aggressor, bystander – are clearly differentiated. In the present study, in contrast, the labels of "victim" and "bystander" were applied by labeling the first individual to show distress following a known or unknown stimulus as the "victim", while all nearby individuals were labeled "bystanders." In our study, temporal differences between displays of distress were rather clear within these managed groups, with the bystanders responding after the victim's first display of distress. This makes it unlikely that these responses concerned the same stimulus, and suggests that they rather concerned the other's distress. If so, the observed behavior is to be interpreted in the same way as consolation in primates, including chimpanzees (*Romero, Castellanos & de Waal, 2010*; *Romero & de Waal, 2010*). Since our study shows that, across distressed individuals, bystanders initiated affiliative contact more often than did victims, the observed reactions seem similar to what is sometimes called "true consolation" in nonhuman primates (*de Waal & Aureli, 1996*; *Romero, Castellanos & de Waal, 2010*; *Romero & de Waal, 2010*).

In studies of consolation, the matching of another's emotional state through emotional contagion (*Hatfield, Cacioppo & Rapson, 1994*; *de Waal, 2003*) may imply that the behavior has empathic underpinnings (*Preston & de Waal, 2002*). In our study, the emotional response of multiple individuals to mostly unknown stimuli could be either contagious (multiple individuals adopt the emotional state of one) or universal (all individuals react with similar emotion to the same stimulus). Substantial anecdotal

evidence of emotional contagion in elephants (e.g., *Douglas-Hamilton & Douglas-Hamilton, 1975*; *Moss, 1988*; *Poole, 1996*; *Schulte, 2000*; *Payne, 2003*; *Bates et al., 2008*) suggests the presence of the required capacity, and the aforementioned temporal differences between the responses of victims and bystanders suggests emotional contagion in this study as well. However, we acknowledge that both interpretations are possible.

This is the first systematic assessment of post-distress affiliative behavior in elephants, and this captive population provided sufficient opportunities to observe this species' capacity for reassurance. We recognize that wild elephant studies often delineate "family" subgroups from "non-family" subgroups through descriptions of affiliative behavior, and so we avoided this terminology since the origin of our subgroups is different. Instead, we focus solely on assessing the capacity for reassurance behavior and analyze how it is exhibited between individuals. The confounds of elephant captivity did preclude us, however, from assessing how this behavior varies with the quality of the relationship between individuals. The use of captive elephants provides an opportunity to investigate whether or not elephants have the capacity for the same level of affiliative behavior following distress seen in consolatory species like the great apes. Future studies on wild elephants should confirm these results and those presented in anecdotal reports (e.g., *Douglas-Hamilton et al., 2006*; *Bates et al., 2008*; *Hart, Hart & Pinter-Wollman, 2008*), even though limitations exist on wild Asian elephant social observations (e.g., dense forest cover and the rarity of consistent, large family group sightings – *Lair, 1997*; *Sukumar, 2006*; *de Silva, Ranjeewa & Kryazhimskiy, 2011*). After all, the original studies of consolation in non-human primates were conducted on captive animals (e.g., *de Waal & van Roosmalen, 1979*; *de Waal & Aureli, 1996*) and were confirmed only much later in the wild (e.g., *Wittig & Boesch, 2003*; *Kutsukake & Castles, 2004*).

This study of post-distress behavior is unique in that it goes beyond the traditional attention to physical contact. The consistent use of vocalizations by bystanders to distressed companions may serve to reassure them, perhaps independent of or to complement physical touches. Both the overwhelming number of unsolicited contacts, and the prevalence of specific vocalizations (e.g., chirping, which may serve as a reassurance vocalization used when elephants are in close proximity to each other – *Nair et al., 2009*; *de Silva, 2010*) lend support to the notion that elephants use multiple communicative modalities (tactile and acoustic) in their affiliative interactions with others (e.g., *Langbauer, 2000*; *McComb et al., 2000*; *Douglas-Hamilton et al., 2006*; *Bates et al., 2008*). In addition, a bystander often affiliated physically with a distressed individual by touching or putting its trunk inside the victim's mouth. This may mirror similar vulnerable contact behavior seen in chimpanzees, whereby individuals put a finger or a hand into the mouth of a distressed other (*de Waal, 1982*, *1990*; *Nishida et al., 2010*).

Bystander affiliation directed toward others in distress, either in the form of consolation following conflict or reassurance following another stressful event, is rare in the animal kingdom possibly due to the unique cognitive mechanisms that may underlie it. Similarities in the complexity of chimpanzee and elephant social relationships (*de Waal, 1982*; *Payne, 2003*; *Plotnik, de Waal & Reiss, 2006*; *Bates et al., 2008*; *de Waal, 2008*;

Byrne et al., 2009; de Waal, 2009; de Silva, Ranjeewa & Kryazhimskiy, 2011; Moss, Croze & Lee, 2011; Plotnik et al., 2011) suggest convergent cognitive evolution that should be further explored through careful analysis of social networks (de Silva, Ranjeewa & Kryazhimskiy, 2011) and these species' use of multi-modal communication in negotiating their physical and social environments.

## ACKNOWLEDGEMENTS

We thank P Panyawattanaporn and A Vijit of the National Research Council of Thailand for their support. We thank T Romero, P Lee, L Rogers and an anonymous reviewer for comments on an earlier version of this manuscript. We also thank S Chailert for allowing us to study the Elephant Nature Park's elephants, and J Smith for his support in organizing the project. We are grateful to K Cullen, G Hayworth, G Highet, J Hilton, M Kobylka, J Schurer, and J Thomas for field support, and the more than forty mahouts – elephant caretakers – who were a constant source of protection and encouragement. This paper is dedicated to the memory of Karl Cullen, one of the world's greatest elephant caretakers, without whom this study could not have been completed.

### Funding

JMP was supported by a US Department of Education Fulbright-Hays Doctoral Dissertation Research Abroad Fellowship. This work was funded in part by the Living Links Center of the Yerkes National Primate Research Center, and the Laney Graduate School of Emory University. The funders had no role in study design, data collection and analysis, decision to publish, or preparation of the manuscript.

### Grant Disclosures

The following grant information was disclosed by the authors:
Living Links Center of the Yerkes National Primate Research Center
Laney Graduate School of Emory University

### Competing Interests

JMP is the CEO of Think Elephants International, a U.S. non-profit public charity 501(c)3 based in New York and currently working in Thailand and elsewhere to link elephant behavior and intelligence research with conservation education. Otherwise, the authors declare no Competing Interests.

### Author Contributions

- Joshua M. Plotnik conceived and designed the experiments, performed the experiments, analyzed the data, contributed reagents/materials/analysis tools, wrote the paper, prepared figures and/or tables, reviewed drafts of the paper.
- Frans B.M. de Waal conceived and designed the experiments, contributed reagents/materials/analysis tools, wrote the paper, reviewed drafts of the paper.

## Animal Ethics

The following information was supplied relating to ethical approvals (i.e., approving body and any reference numbers):

This project was approved by the National Research Council of Thailand and Emory University's Institutional Animal Care and Use Committee (ID 219-2007Y).

## Field Study Permissions

The following information was supplied relating to ethical approvals (i.e., approving body and any reference numbers):

National Research Council of Thailand.

## Supplemental Information

Supplemental information for this article can be found online at
http://dx.doi.org/10.7717/peerj.278.

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
