# Peer review of "Asian elephants (Elephas maximus) reassure others in distress"

_PeerJ, doi:10.7717/peerj.278_

## Round 0.1 · original submission · Minor Revisions

Please take all of the comments make by the reviewers into account in your revision.

·

Basic reporting

The main suggestions I have for improving this paper are associated with the writing style, which needs to be smoothed out, particularly in the abstract and introduction. I have highlighted and commented on some examples of this in the paper attached.

I suspect that there may be some over-use of self-citation, and in some cases over-referencing.

Use of bracketed information is somewhat excessive, as is use of "i.e." and "e.g.". This interrupts the flow of the writing, and the points could be incorporated as succinct sentences in their own right. Use of "c.f." in referencing could also be reduced.

In some instances redundant wording has been used and could be removed (more editing needed).

Otherwise this is a very interesting scientific paper, and the authors have used suitable methodology and explained their results clearly.

Experimental design

No comments

Validity of the findings

No comments

·

Basic reporting

Generally good - I have a few comments for authors where clarification of reporting would be helpful and support their findings.

Experimental design

I have concerns about the use of the term pseudo-family for these artificially constructed units in a captive environment. In captive elephants, small units of associates may form due to the nature of space constraints, but these may be lacking the friendly social interactions which characterise families. Among wild African elephants, we define families on the basis of their supportive reassurance vocalisations and contacts during stress events, which make these definitions and tests circular. In the wild elephants, we now have genetics and generational genealogies to support the kin-based nature of most families; some however are simply affiliative units made up of females who have chosen to be in close proximity to each other, which then enable persisting supportive interactions. So while I would agree that non-kin can be supportive of their other group members (as can adult males), I would like to see a better justification of the definitions of these small groups – how can this be done if the nature of a family is precisely the reassurance and support that is being tested here?
Is the 10 minute PD sample relevant to elephants? Do they react more quickly and calm less easily than primates?
Is “bunching” defined or referenced? (not until line 296). McComb has a lovely designation of bunching as a stress response.

Validity of the findings

Generally robust data and analyses - a few places where clarifications could be helpful.

Additional comments

I am actually very supportive of these observations – of course elephants “do” reassurance (and not just of infants, as noted above, adult males will exchange help/support against external risks). My problem is with the circularity of the construct. Pseudo families show significant reassurance after a stressor; how is this different from the definition of the pseudo family?
You demonstrate contagion among the friends when one individual exhibits distress – this also is a common feature of families.
Another interesting question is that distress is communicated widely through vocal signals – all elephants whether close or far are probably aware of the distress event and responses (which if escalated can cause further distress again due to contagion). My point here is that if only "friends" respond to a widely dispersed signal, then it may indeed be "special".

I am not convinced that “vocalisations signal agitation or excitement” – in my opinion that is the result of captivity. Most vocalisations in the wild are movement signals (“let’s go”), affilitaive reassurance after a brief separation, play signals, friendly “greetings” – the most common. So, Olson’s work relates really to stressed and socially deprived elephants and should not be used as a model.
Urination / deification + vocalisation is a signal of arousal, associated with a series of loud rumbles, roars, bellows, screams, trumpets (or chirps in Asian elephants?). These are quite distinct from the greetings, play and movement vocalisations.
“bystanders responding with a considerable delay following the victim’s first display of distress” – these results seem sound, but what is a considerable delay? These results are not presented.

---

## Round 0.2 · accepted · Accept

I suggest that you might move the first paragraph of the Results (lines 273-287) to the Methods.